# Neighborhood Built and Social Environment Influences on Lifestyle Behaviors among College Students in a High-Density City: A Photovoice Study

**DOI:** 10.3390/ijerph192416558

**Published:** 2022-12-09

**Authors:** Ming Yu Claudia Wong, Kailing Ou, Chun-Qing Zhang, Ru Zhang

**Affiliations:** 1Department of Health and Physical Education, The Education University of Hong Kong, Hong Kong SAR, China; 2Department of Sport, Physical Education and Health, Hong Kong Baptist University, Hong Kong SAR, China; 3Department of Psychology, Sun Yat-sen University, Guangzhou 510006, China; 4School of Physical Education & Sports Science, South China Normal University, Guangzhou 510006, China

**Keywords:** physical activity, photovoice, eating habits, active transportation, physical environment

## Abstract

Based on the social ecological approach, a photovoice study was conducted to explore how neighborhood built and social environments facilitate or hinder college students’ lifestyle behaviors, including physical activity, active transportation, and dietary behavior. A total of 37 college students took photos about neighborhood built and social environments that may affect their physical activity, dietary behavior, and active transportation, and shared their perceptions about how neighborhood built and social environments influence their lifestyle behaviors. Our findings demonstrated that the availability and accessibility of services, school facilities, and home facilities affected physical activity and dietary behaviors among college students. Moreover, the well-developed transportation facilities and networks benefit college students’ active transportation. Environments-based interventions are recommended in future research to better understand the associations between neighborhood built and social environments and lifestyle behaviors in college students.

## 1. Introduction

Social environment disparities were marked as an important factor in restricting the mobility of communities; these disparities have worsened as a result of the current global health catastrophe bought on by the coronavirus [1]. In Pipitone and Jović (2021)’s study [1], they indicated the importance of urban green spaces in enhancing people’s physical activity, emotions, and social cohesion. College students are a group of young adults who are in a transit period from adolescence to adulthood [2]. The benefits of healthy lifestyle behaviors (e.g., regular physical activity and healthy dietary behavior) during young adulthood can extend to later life and make individuals achieve better physical health and mental well-being [3,4,5]. To promote healthy lifestyles among college students, a deep and systematic understanding of lifestyle correlates is required [6]. According to the social ecological model [7], healthy lifestyle behaviors are highly influenced by interpersonal support, physical environment, social environment, transportation environment, as well as policy or community market condition. These environments were also referred to as activity spaces [8]. For example, an activity space designates the locations and spaces an individual interacts with, thus increasing their engagement in activities, thereby providing a dynamic measure of the environment [9]. Despite these activity spaces being seen as intimate places, like home and work locations, they also included food outlets, schools of students, open spaces near home or school, as well as social environment where people meet their friends [10]. Previous studies revealed that a healthy lifestyle of college students is affected by the environmental feature, including the walkability of the sideways, safety, and the availability of green spaces [11,12]. Moreover, the social network environment, including peers and the available electronic physical activity platform, was shown to facilitate college students’ physical activity participation [13,14]. While, the social ecological model provides a useful theoretical guidance on understanding the college students’ healthy lifestyle behaviors [6]

Regarding the influences of environmental factors on college students’ physical activity, reviews indicated that people’s moderate-to-vigorous physical activity and walking rates were higher in activity spaces that were more walkable, more densely populated, greenery, or had utilitarian services [15,16,17]. However, the environmental influences on college students’ physical activity were variated, which might be due to the differences in green spaces, streets densities, peers, and social network [8]. For example, research found that built environment features within 1 km and 2 km of residential and school areas, such as recreation facilities, parks, walkable pedestrians, showed a positive influence on enhancing physical activity [18,19]. Other than open spaces like parks, pedestrian collision was also seen as a risk factor in affecting the participants’ level of walking [20]. For example, a recent study on college students found that traffic and pedestrian collision near the college campus mostly affected students walkability [21]. However, most of the previous findings were revealed in low density urban and rural places, while there is a lack of studies on the high-density metropolis, which is very important given the growing number of big cities worldwide. 

Transportation behavior is also treated as a predictor of healthy behaviors among college students [22]. Yet, transportation mode is highly restricted by the public physical environment, such as school proximity, the accessibility to services (e.g., clinics, recreation activities centers, rail, or bus station), walkability of pedestrian or footbridge, as well as the weather climate [23]. As such, school students, including college students, tended to engage in a healthy travel mode, walking, were those living near their schools, as well as having a higher accessibility to public transport stations [23]. A study conducted in Australia evidenced that college students’ transport-related cycling is associated with the traffic environment, as well as personal motives and barriers [24]. Additionally, novel cartographic mapping research has demonstrated that the magnitude and spatiality of environmental factors, including density, diversity, design, and household across the college’s region, would critically influence college students’ active transportation; yet distance was not considered a common barrier [25]. For college students to live in the campuses located in high-density cities, like Hong Kong, there might not be enough dormitories. Accordingly, they have to rely on private or public transportation on their daily commutes and other recreational purposes to reach distanced places. However, there is a scarce of studies exploring the influences of environmental factors on the transportation behaviors among college students in high-density cities where they live in their own homes or rented apartments. 

Beyond the influences of built environments, the social environment has also been demonstrated as a key factor on cultivating healthy lifestyle behaviors among college students [26]. The dimensions of social environment, including social support, social network, social cohesion, and neighborhood factors, were described as essential factors in predicting people’s healthy behaviors, especially physical activity and dietary habit [27]. Dietary habit, in particular, the social environment, like peers and family members, were shown as significant networks that impact individuals’ food choices, energy intake, and eating habits. As compared to students who ate alone, with friends or other people, those who ate with their families adopted a more healthy and balanced diet, with less energy-dense foods and beverages intake [28]. That said, college students are at a critical stage of growing into complete independence, and social environments play a key role in their lifestyles. 

Different environments and activity spaces can influence college students’ healthy lifestyle behaviors. In Hong Kong, studies targeting students in a variety of activity spaces, including the social, home, market, transportation, and school environments, have not been conducted to determine their impact on healthy lifestyle behaviors [29,30]. The social ecological model [31], which has emphasized the dynamicity, inter-relational, and multifaceted relationships between individuals and the environment through personal, social environment, physical environment, and policy status of the community. Extending this well-suited framework for multilevel environmental influences on college students’ healthy lifestyle behaviors is a key for future intervention studies [31]. On the other hand, there is a lack of empirical studies on exploring the environmental influences on healthy lifestyles of college students in a high population density metropolis. Using Hong Kong as an example, the current study therefore aimed to qualitatively explore how different elements within the home, social, and physical environments help or hinder college students to be physically active in recreation, and to consume healthy food (i.e., eating fruits and vegetables, and lower consumption of sugar-sweetened beverages) by asking college students to share their stories associated with each of their most meaningful photographs (photo-stories).

## 2. Methods

### 2.1. Participants and Study Methodology 

A convenience sample of 37 college students (70% of female, *M*_age_ = 19.95 *SD* = 1.10, age range: 18–23) was recruited via distributing leaflets in a public college in Hong Kong to participate in the photovoice study. The photovoice method requires participants to take photographs of the investigating topic and the photography would act as information pieces to guide the in-depth interviews [32]. This is commonly used in understanding community development and public health behaviors. With the photovoice approach, people are able to recognize, represent, and enhance their communities through specific photographic techniques [33], in order to obtain a deep and bottom-up understanding of health. By using cameras, people can photograph their perceived work and health realities. Photovoice was used in this study to allow participants to record and reflect on their personal and community environment regarding their healthy lifestyle behaviors, as well as their daily lives [33]. Therefore, in the initial in-person meetings, participants were asked to take photos to record how participants think about the influences of the environments on their healthy lifestyle behaviors in terms of promoting or hindering them to engage in physical activity, and to consume healthy food for two weeks. Using these photos taken by participants, they were again invited to engage in an in-depth interview. The in-depth interview was conducted by the fourth author, RZ, focusing on exploring and investigating the content of the photos, and how these environments reflected by the photos affect their lifestyle behaviors, including doing physical activity, transportation, and food intake behavior. The Committee on the Use of Human and Animal Subjects in Teaching and Research (HASC) from Hong Kong Baptist University approved the current study (No. FRG1/17-18/031).

### 2.2. Procedures

Prior to the start of data collection, an initial meeting was held to introduce the participants to the underlying issues about the use of cameras, power and ethics. The concept of photovoice was also introduced to participants, along with the respective purpose of encouraging participants to understand the community environment, appreciating their home and school environment for healthy eating and exercising habits. These procedures were used to assure that participants could fully understand what kinds of photographs they should take, which included the physical environments of their homes, schools, neighborhoods, and aspects of policy and economy related to their healthy lifestyles. Based on the Social-Ecological Model and previous studies [31,34,35], the following categories were used and introduced to participants to seek for: 

Dietary behavior, including: (a) Marketing and policy: price/affordability, online shopping, advertisement; (b) Physical environment: availability and accessibility of food stores, supermarkets; healthy or unhealthy options in food outlets, vending tool; (c) Social environment: peer and parental influence, with whom college students eating together. Cultural preferences, such as cooking themselves; (d) School environment: canteen (hygiene and sanitation, price, availability of food), vending machine, and coffee shop; and (e) Home environment: food availability and diversity at home, family rules on food.

Recreational physical activity, including: (a) Marketing and policy: information and provision of community-based physical activity program, affordability of recreational facilities and active clubs, availability of free physical activity programs, fundraising to support physical activity, physical education policy at universities, and after school physical activity programs; (b) Physical environment: sidewalks, walking trails, recreational facilities, distance from home to active places, traffic safety, street lighting, aesthetics of built environment, school active places, trash, street lighting; (c) Social environment: crime safety, social influence of parents and peers; (d) Home environment: sport equipment, indoor activities; and (e) Influences of climate and air pollution.

Active transportation, including: (a) Marketing and policy: affordability of public transportation, government financial support of active transportation, convenience of public transportation; (b) Physical environment: sidewalks, walking trails, distance, traffic safety, aesthetics of street, street lighting, walking facilities, and pedestrian sign; (c) Social environment: crime safety, social influence of parents and peers; and (d) Influences of climate and air pollution.

Informed consents were firstly obtained from participants after the purpose of this study was explained to them. Participants were informed that they had the freedom to choose what, when, where, and how many photographs to take. Participants were also reminded to seek approval when other persons are involved in the picture. After the initial meeting, participants were allowed to use two weeks’ time to take photos that they felt facilitated or hindered their physical activity and food intake behavior. There was no defined minimum or maximum number of photographs that should be taken, but participants were advised that in the follow-up interview, there would only be time to discuss a handful of photos. Following two-weeks of photo taking time, participants were invited to have an in-depth interview with the fourth author, RZ, to discuss the photographs taken. 

In the face-to-face in-depth interviews, each participant was asked to select and talk about one or two photographs of each healthy lifestyle behavior that the student they believed are most significant and/or that are their favorited photos. Sampled interview questions are: (a) What do you see here? (b) What is really happening here in this picture? (c) How does this relate to your physical activity or eating habits? (d) How does school/family/society have an influence on your behavior? and (e) The interviewers have at least gone through photos for each of the behaviors (food, physical activity, and transportation). The in-depth interviews were audio recorded with participants’ permission and they lasted from 45 to 60 min. The interviews were conducted in Chinese and the audio recordings were transformed into texts verbatim. Participants were invited to check the accuracy of the interview texts for accuracy.

### 2.3. Data Analysis

In line with previous photovoice studies [35,36,37], the inductive and deductive approaches for thematic analysis were adopted to analyze the qualitative data using the MAXQDA 2022 Software (https://www.maxqda.com/new-maxqda-2022) [38]. First, audio files were transcribed verbatim and any identifying data were removed and replaced with code names. Second, the coders (MYCW and KLO) first read and re-read the transcribed data to ensure familiarity. The analysis begun with the inductive approach, all extracts with the dataset were coded into meaning units, and then were resorted and collated according to potential themes capturing salient patterns in the data. The themes were reviewed to ensure they were consistent with the original interview transcripts. The terms from the data were used to assign the names of the themes where possible. Then, at the stage of using the deductive approach, higher-order themes were formed into different dimensions based on the theoretical guidance of applying the Social Ecological Model of healthy lifestyle behaviors in school environments [37]. The coding was conducted by two co-authors (MYCW and KLO) independently. The coding results of one coder (e.g., MYCW) were checked and refined by another coder (e.g., KLO) to ensure the coding stability on tags and themes. Any inconsistent coding was discussed by the two coders, with discrepancies resolved after mutual agreement. The consolidated criteria for reporting the qualitative (COREQ) research checklist for interviews was used as a guidance for reporting findings of our study [39].

## 3. Results

Findings of this photovoice study on the influences of built and social environmental factors on healthy lifestyle behaviors among college students in Hong Kong can be summarized into three main themes: (a) dietary behavior; (b) recreational physical activity; and (c) transportation and physical activity. Direct quotes from participants were presented and acronyms of pseudo names of participants were used. 

### 3.1. Dietary Behavior

The photos and examples of codes and sub-themes on dietary behaviors among college students are shown in Table 1. It is unsurprisingly that, in our study, college students’ dietary habits tend toward fast food, fried food, dessert, and snacks; they prefer to satisfy their taste buds, although they know they are unhealthy. Marketing policies such as advertising, discounts, student offers, online shopping, and affordable buffet impact most on their food behavior. They would be attracted to novelty and trending snacks and restaurants, even though it is not their first choice sometimes.

*“Because the taste that has been set since childhood, it’s weird to have the sugar-free food. So, whether it is healthy or tasty, I will go for the tasty ones.”* (WCY)

*“In Hong Kong, there’s often publicity for trending restaurants and signature food, so I’m going to queue up to try it. It’s not that I want to go there for dessert, it’s that it’s so famous that I want to go there to take pictures and try it.”* (SUM)

College campus’ catering environments influence students’ healthy dietary habits. Most students complained that the quality of food in their school cafeteria is bad, for instance, less healthy food selection, heavy oil and salt, and poor hygiene, which force them to dine outside. Meanwhile, facing less choice at the canteen, dining out with classmates and friends has become the common social gathering for students, and there are lots of promotions in different restaurants, therefore they would intake unhealthier foods. They also claimed they do not care about healthy food when having a meal with friends. 

*“The drinks at school are all sodas, sweet and high in sugar. And Ovaltine and other things are also sweet and high in calories, which is not healthy for us.”* (WCY)

*“There is less vegetable choice in our school canteen. Once I ordered meat rice and there was only one green vegetable in it.”* (DCY)

*“I think having a dinner party with friends might make you eat a bit more freely... Because you would be very enjoyable and then order a lot of things, so that can be a barrier to healthy eating.”* (WZW)

Family dietary habits may also influence college students’ dietary behaviors. Most participants expressed they tend to intake healthier and lighter foods when they eat at home, because their parents knew they might eat unhealthy meals outside, so their parents would cook healthier and more nutritious dishes such as vegetables, fruits, soups, etc. Meanwhile, students would choose to cook easier and healthier foods when they are at home. However, some family members have unhealthy eating habits that would also affect students, for example, order takeaway food and snacking.

*“The principle of eating in my family is that my mom insistence that we must eat vegetables.”* (DHY)

*“As my father is not in good health, and my mother knows that we all eat fatty food outside, so she would cook lighter at home.”* (WWK)

*“We don’t cook at home; we usually store some noodles and canned goods at home... or takeaway some fast food.”* (SUM)

For physical environment, it was found that there are lots of fast-food restaurants, such as McDonald’s, KFC, and Pizza Hut, located near their neighborhood. Therefore, lots of students choose these restaurants because they are inexpensive and convenient. 


*“This McDonald’s has put its dessert shop outside on the street, which I think is a big obstacle for our health. It’s quite common to pass by and buy an ice cream especially in hot weather like now. And McDonald’s is also cheaper than the others.” (ZZY)*


### 3.2. Recreational Physical Activity

The photos and examples of codes and sub-themes of recreational physical activities among college students are shown in Table 2. Home environment and family support were shown to affect recreational physical activity participation, in particular physical activity. Participants expressed a higher level of interest in participating in physical activity when they have equipment at home, such as a yoga mat and dumbbells, where they could do exercises that require less space, like yoga, sit-ups, push-ups, and arm-curls. Students’ physical activity habits would also be facilitated by recreation facilities in the neighborhoods. These facilities include the estate clubhouse with a gym and pool, nearby parks with running tracks, and a football or basketball court. Additionally, exercising with family members was identified as a motivation for the participants to engage in physical activity. For example, spending time running with siblings, spending time swimming, and playing badminton with mother and father, or heading out and taking a walk in the park after dinner.

*“Yoga mat. I bought it for a long time, I put it at home, and then I can do exercise anytime when I’m free.”* (DHY)

*“These are the two pieces of equipment I have in my home, a yoga mat and a dumbbell. Occasionally, on a whim, I will use a yoga mat to do a little bit of yoga.”* (CL)

*“There are three or four sports fields near my home, including soccer and basketball. I will get out and play when I’m free, because I like sports as well.”* (FYX)

*“The pool in the picture is at my clubhouse. It’s on the fifth floor of our clubhouse. It’s not standard, but it’s not too crowded, and it’s only open to the owners who live there, so it’s quite convenient.”* (FYX)

In addition to the home environment, the built environment is considered to be the second most intimate location for a college student. Students expressed a higher level of intention to engage in physical activity when school sport facilities are accessible. College students who lived in the dormitory showed a higher frequency of campus gym room usage. Students were also interested in joining recreational activities held by the dormitory or school, for instance, the dormitory hiking group, and the inter-dormitory basketball, or soccer team. In contrast, other sport facilities, such as football courts, basketball courts, or tables tennis centers, were considered as limited; therefore, they might consider the public community sports facilities near their dormitory rather than those at the college. The college campus’ facilities would influence their intention to exercise, and some students might choose to work out during low-peak hours, but time is regarded as unpredictable, resulting in unstable workout schedules.

*“The schedule of the school’s fitness room collided with my class schedule, so there is no way to do fitness in our free time, although the other fitness room is not full of schedule, but because the big one is not available, so many people are crowded on this side of the small one, the space is very small.”* (WZY)

*“This is the school gym. I have been to rent that field with my classmates to play badminton.”* (NPY)

*“There are regular activities in the dormitory, and my roommate and I occasionally participate in interhall competitions, such as dodgeball and stickball.”* (MK)

Built environments of campuses facilitated students’ social environment (i.e., social support, social network) for engaging in recreational activities. Participants of our study emphasized that they tended to enjoy cycling or hiking during weekends with friends as leisure activities. They would choose to go exercise at the campus sport center together after class as well. However, physical environments of these sporting facilities could also be a barrier to students for doing physical activity. Other than being crowded, the quality of the machine and facilities (i.e., pristine condition, new) and the hygiene of the sporting area would be considered by the students before deciding to go exercise; along with the weather. When the weather is too hot, too shiny, raining, and humid, students will be less likely to participate in physical activity.

*“Sometimes it will be renovated, you can also see, is rotten, sealed, people can only pass through that side, it is inconvenient.”* (YWS)

*“There will be too many cars if I run here, because there is no running trail near my home, so I can only run near the road, and there are too many cars. Interviewer: Not safe? Interviewee: Yes, there are too many cars, the air quality is low.”* (WWY)

*“Kowloon Bay sports ground, soccer field, stadium, and park are all close to each other, but the surrounding geographical environment is very remote. It is built in a factory area, and it is different from the usual sports ground, so most people need to take a transport bus to get there. Also, there were very few people walking around and the lights were very dim. The Kowloon Bay Sports Complex has only one floor and no gym, so sometimes there is no chance to go to a more remote place for fitness.”* (WZY)

In their activity spaces, participants of this study also mentioned some positive sides of physical environments that facilitate their physical activity. For example, sport electronic devices, such as smart watches or apps that can record physical activity or walking performances, can help facilitate self-monitoring and self-determination in the pursuit of regular and consistent physical activity.

*“It’s about a five-minute walk from my house. Then it has all kinds of facilities, a gym, and some courts.”* (HWY)

*“As you can see, there are lots of stools here. Because there are many residential buildings around... the weather will be cool at around seven o’clock in the evening, people will play sports or rest here, and it is very convenient.”* (YXY) 

*“Using sport electronic devices allows me to understand what my heart rate is like when I’m doing exercise and how many calories I can burn. It gives me more insight into what kind of a result I can get from this exercise, so I like that.”* (WZW)

### 3.3. Transportation and Physical Activity

The photos and examples of codes and sub-themes on transportation and physical activities among college students are shown in Table 3. Participants expressed a consistent perception towards the convenience of Hong Kong transportation, where metro stations and bus stations, the most common public transportation in Hong Kong, are highly accessible. In other words, the ease of access to public transportation reduced the intention of walking as a mode of transportation. Furthermore, the physical environment of Hong Kong would also lead them to take public transport instead of walking. For example, metro stations could prevent traffic jams so they could be on time for any events. Additionally, participants may choose to take public transportation late at night to ensure safety. In addition, the distance and the weather (such as during the summer or a rainy day) would also influence their decision to travel by public transportation.

*“Because the subway is more punctual and will not be jammed, the time is easier to control.”* (FXY)

*“It is very dark and dangerous on the way from the dormitory to Kowloon City... Usually, I don’t walk alone on this road.”* (HWC)

It should be noted that, most of the participants in our study were living near the college campus when the interviews were conducted. Therefore, walking is clearly that first choice. Students tended to walk due to the expensive price of taking public transport, in particular, the minibus near the campus.

*“Because I have to pay for the whole trip, if I want to take the car to school, I have to pay nearly six dollars; then if it’s a round trip, I have to pay almost 12 dollars a day for the care fare, I think it’s very uneconomical, so I’ll choose to walk.”* (DHY)

*“No, I mainly consider the fare, that is to say, if I travel by public transport every day, the daily fare will be ten dollars; But if I choose to walk, I save this money and use it for other purposes.”* (LMF)

In addition, the condition and environment of the street were also seen as contributing factors. The participants expressed those roads with a cover (e.g., covered footbridges), street lights, green spaces, and aesthetically pleasing roads would encourage them to walk as a mode of transportation.

Furthermore, apart from walking, cycling is also a form of healthy travel mode. Shared bicycles were evolving in the mainland China community, but not in Hong Kong. Participants expressed that those shared bicycles were expensive, and they found it difficult to use cycling as a travelling mode in Hong Kong. Cycling is not allowed on pedestrian walkways, but it is relatively dangerous to ride on the road. Therefore, cycling cannot be considered an effective healthy mode of transportation in Hong Kong. There were only a few interviewees who indicated that they might use cycling as a transportation mode sometimes. In common with all of the interviewees is that they are all living in the New Territories, like Sha Tin, Hang Hau, where there are cycling tracks that allow them to travel within short distances by bicycle. It is because there are only cycling tracks in the New Territories but not in other areas of Hong Kong.

*“This is where I would ride my bike when I had to tutor. Yes, because the kid I was tutoring lived in Hang Hau, so it was very close (Interviewer: There is also a bike lane?) It’s safe to have a bike lane.”* (NPY)

Please note that Figure 1, Figure 2 and Figure 3 have displayed the coding tress of each theme.

## 4. Discussion

Using the photovoice study approach of photos of everyday life followed by in-depth interviews, the current study has explored the bottom-up perceptions of the healthy lifestyle behaviors among Hong Kong college students. Findings on students’ perceived influences of environmental factors on healthy lifestyle behaviors can be summarized into the main themes of dietary behavior, recreational physical activity, and transportation and physical activity. It seems that Hong Kong college students’ healthy lifestyle behaviors, especially physical activity participation and dietary habits, varied according to the built and social environments. Findings of the current study can be applied into the macro levels of cultural preferences, marketing and policy, the meso levels of physical and social environments, for example, the school and home environments.

Findings of this study showed that as college students are in the transition phase from living at home to living alone or with peers, many food choices are deeply involved in this change. Based on the Social Ecological Model [31], college students’ eating habits are affected by their healthy and unhealthy eating patterns, together with social environment (family and friends), as well as barriers and enabling environments (e.g., restaurant access, school canteen) and market policies (e.g., students discount, promotion) [7]. In accordance with many researchers [30,40,41], most college students have unhealthy eating habits, especially due to their taste preferences, such as eating fast food, eating irregular meals (e.g., bedtime snacks), and consuming unhealthy snacks, desserts, and other junk food (e.g., fast food, beverages). Moreover, students’ perceived benefits of healthy eating also influence the intention to consume healthier food [42] because self-control, health motives, and self-risk of unhealthy eating play a significant role in healthy eating. To change students’ eating habits, lots of evidence [41,43,44] suggests that increasing students’ nutrition literacy by providing a tailor nutrition-related intervention, such as knowledge of consumption of fruits and vegetables, using technology in education, cooking practices, as well as weight managements. 

The important role of social environment on college students’ dietary behaviors was supported in the current study. Young adults’ food intake was also moderated by their identified norm reference group, because they are in a period of life in which desires to be liked and ‘fit in’ to peer groups are high [45]. Therefore, there is potential to design social norm intervention by using social norm messages to encourage healthier eating, for instance, selecting lower calorie options, smaller portions sizes, or plant-based meals [46,47]. Findings of our study highlight that college students were influenced by the rich but unhealthy food environments in Hong Kong. For food environments, easier accessible fast-food restaurants, advertisement of trending snacks, and more affordable “junk food” were barriers for Hong Kong college students to maintain healthy dietary behaviors. Restaurant merchants should consider reformulation of foods, with less sugar, oil, and salt, or replace unhealthy flavoring agents with other healthy raw materials [44]. In addition, interventions across campus dining facilities were expected to reduce potential barriers to healthy food consumption and increase self-efficacy and behavioral control among students, for example, by improving accessibility, availability, and presentation of healthy food items (i.e., whole fruit, fruit salad, vegetarian daily specials) and fostering students’ perceptions of confidence to consume a healthy diet [48].

Findings of the current study revealed that social support such as family, friend, and peer support has strongly affected college students’ physical activity behaviors. College students received higher levels of esteem support from their family, and companionship support from their friends influenced their physical activity engagement [49]. Photos regarding a positive home environment and family recreation activities were recorded more by female than male participants in this study; this can be explained by studies that females were more likely to engage in physical activity if they received family support [50,51]. Peer support also matters to college students. For example, a systematic review claimed that male college students valued social support more from their friends [52].

Findings of the current study further demonstrated that physical environments at home and on campus also influence physical activities of college students. In line with previous review, findings of this study showed that having exercise equipment at home, such as dumbbells and a sport mat, or sports devices, such as an exergaming bike and smart watches, could increase college students’ physical active behaviors [53]. However, students complained there were inadequate open spaces and recreation facilities in the community, as in Hong Kong, open space constitutes only about 2% of the city’s area [54], and according to the standards for the provision of leisure facilities, for example, one badminton court is shared by 8000 people [55]. To benefit students’ facilities usage, it is suggested that the Government could introduce a policy of priority booking of leisure facilities for students or educational groups. What is more, to maximize land use, schools could consider building more multi-purpose sports fields and making reasonable use of outdoor lawns. In addition, urban planners should expand the open space of residential buildings, for example, by using the floors of residential buildings as multifunctional sports courts and by building more paths to parks near residential buildings in rural areas to improve physical activities for local residents (Legislative Council Panel on Home Affairs, 2021).

Findings of the current study showed that, in a high population density metropolis like Hong Kong, the most common transport modes for college students were walking and public transport. This finding is in line with a previous study that revealed Hong Kong adolescents and young people in healthy and sustainable travel modes [23]. According to the market policy in Hong Kong, the MTR student travel scheme offers MTR concessionary fares to eligible full time day course students studying in a recognized institution in Hong Kong to benefit their commuting to school and encourage them to participate in more recreation programs after school [56]. Therefore, most students used MTR for their school commute. In addition, students prefer walking if their school/home is close to public transit stops, although previous studies have suggested that parents can increase their active transport mode for their children (secondary school and below) by choosing the nearest school to attend [23]; this situation does not apply to college students, so they may experience longer school commutes. 

Based on the findings of our study, it is suggested that urban transportation constructors should consider the distance from public transit stops to school, therefore setting an acceptable walking distance from public transit stops to campus can motivate students to walk. Because of the hot climate condition in Hong Kong, it was reported that the threshold for Hong Kong students’ active school transport was 400 m for street greenness and the number of parks surrounding school or 800 m for overall greenness surrounding the school [29]. In addition, changing the built environment, for instance, a greater pedestrian networking connection, increasing recreational budding and street greenness, increasing population density, as well as having more bus services were associated with a higher level of active transport [57]. It is worth noting that, the social or macro aspect discussion using the Social Ecological Model has supported the illustration of the hermeneutic circle based on the interview outcomes. Through the exploration of the impact of the social and neighborhood environment on college students’ healthy lifestyle behaviors, the current study has also reflected on the ongoing circular movement between personal lifestyle and the social phenomenon of Hong Kong (i.e., the food culture, the dense transportation, and urban system). The hermeneutic circle reflects the origin of Hong Kong college students’ sedentary and unhealthy eating habits, and how dense urban development contributes to less active transportation and leisure physical activity. 

The strength of the current study includes using the photovoice approach. This is expected to create a more comprehensive and more dimensional view of Hong Kong college students’ physical, eating, and transportation habits. Participants were able to identify their healthy lifestyle behaviors in different social environments. Nevertheless, the process of taking pictures and recalling the events shown in the pictures has also encouraged Hong Kong college students to become self-aware of their lifestyle behavior and to admit that it is not healthy. In addition to individual responsibility, Hong Kong’s market and environment are considered potential barriers to people’s healthy lifestyle choices. 

Limitations of the current study should be acknowledged. First, a convenience sample was used in the current study. Future research with representative samples is recommended to better examine how neighborhood built and social environments influence lifestyle behaviors in college students. Second, feedback was difficult to seek from the participants due to graduation and loss of contact, hence the extent of data saturation might be affected. Nevertheless, this study also provides themes of health-related behaviors among young adults. It is suggested that future studies involve more students from different backgrounds and use these themes to explore more health behaviors and their correlation with health among college students. Third, the current study did not conduct a pilot study to test the interview questions or include pre-post questions. Fourth, we did not make field notes during or after the interview, suggesting a lack of contextual information. Future research is recommended to conduct field notes during and after the interview to record contextual information. Lastly, we acknowledge a relatively small sample size was used in the current study. Although a large range of sample size from five to fifty was used in the photovoice research, we have to admit that a large sample size would make the research outcomes more persuasive and influential. 

## 5. Conclusions

Using the photovoice approach, the current study explored the influences of different elements within the home, social, and physical environments on college students’ physical activities, dietary behaviors, and active transportation. Findings indicated that college students’ dietary and exercise habits were highly influenced by the physical environment of college campuses and home and neighborhood food environments. College students’ dietary choices and physical activity levels depend on the affordable accessibility of food environments and exercise facilities at the neighborhoods of their college campuses and homes. Active transportation is common among college students in Hong Kong, with college students who live within walking distance of college campuses and choose to walk. Implications of our study are that future intervention endeavors should focus on promoting positive eating and exercise habits among college students through environments-based interventions and policy making from college administration and local government.

## Figures and Tables

**Figure 1 ijerph-19-16558-f001:**
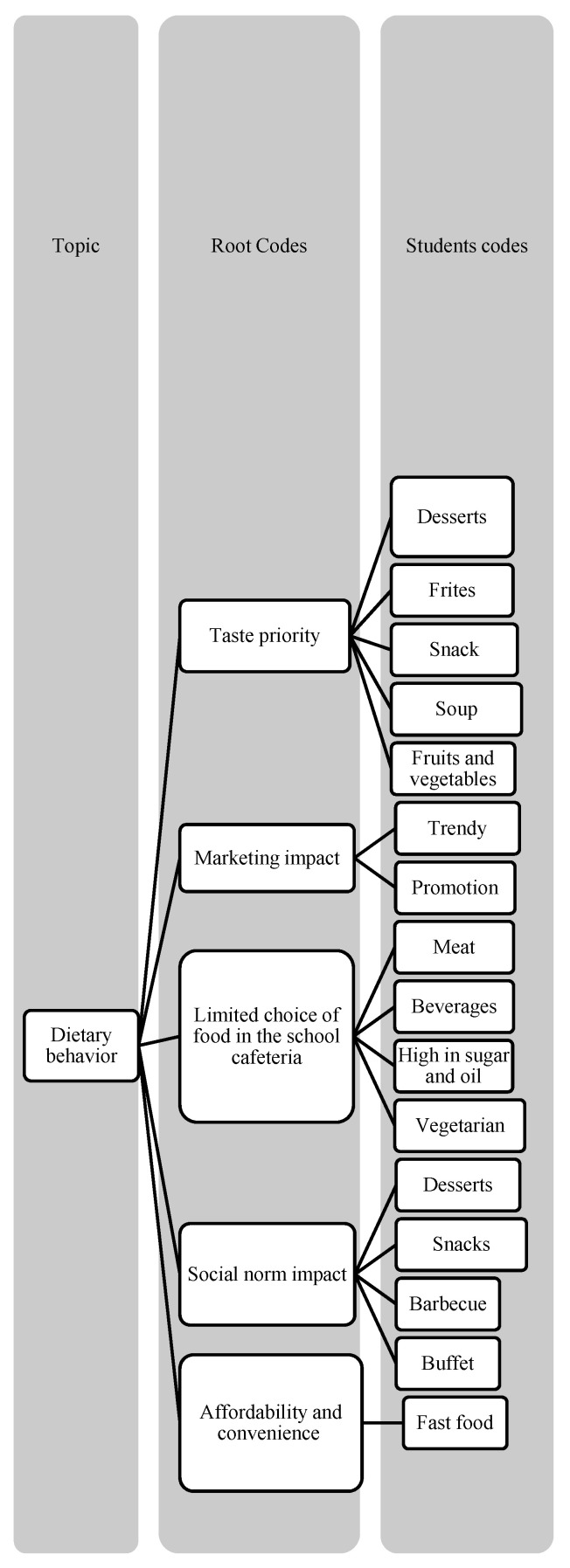
Root codes and students’ codes of dietary behavior.

**Figure 2 ijerph-19-16558-f002:**
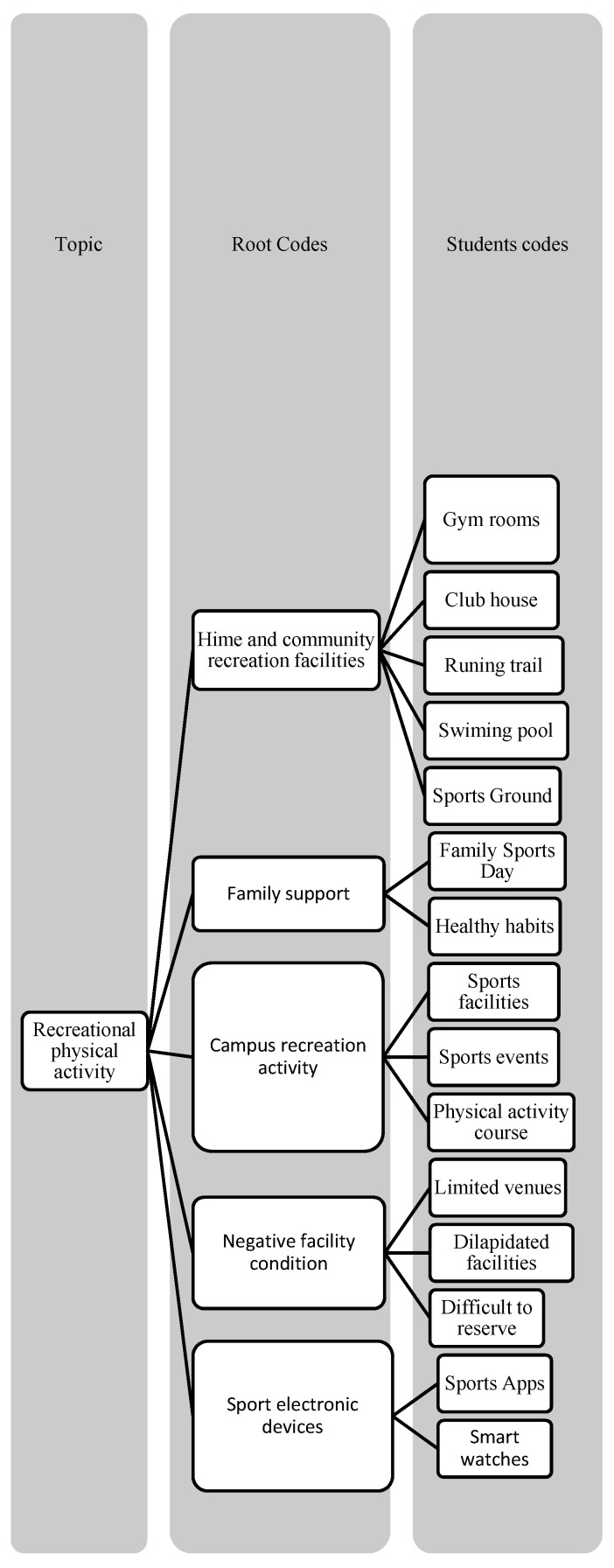
Root codes and students’ codes of recreational physical activity.

**Figure 3 ijerph-19-16558-f003:**
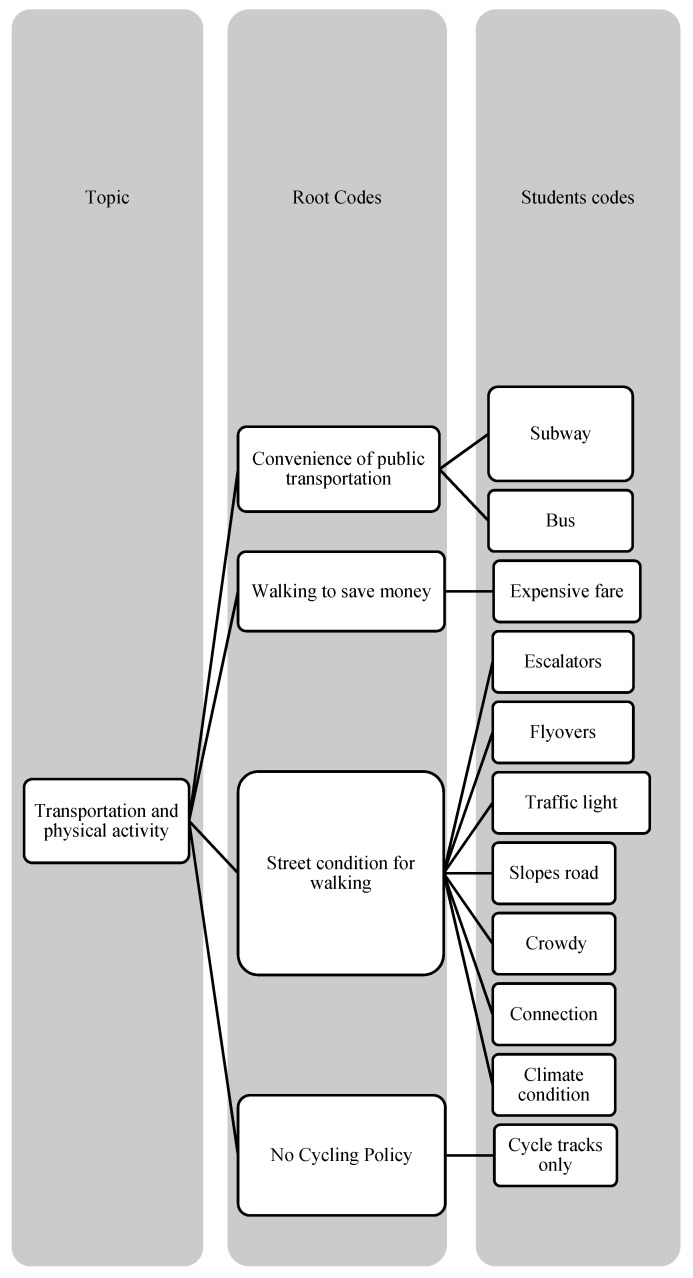
Root codes and students’ codes of transportation and physical activity.

**Table 1 ijerph-19-16558-t001:** Examples of codes and sub-themes on dietary behaviors among university students.

Subthemes	Meaning Units (Codes)
Taste priority	“Because the taste that has been set since childhood, it’s weird to choose the sugar-free food. So, whether it is healthy or tasty, I will go for the tasty ones.” (WCY)
Marketing impact	“In Hong Kong, there’s often publicity for trending restaurants and signature food, so I’m going to queue up to try it. It’s not that I want to go there for dessert, it’s that it’s so famous that I want to go there to take pictures and try it.” (SUM) 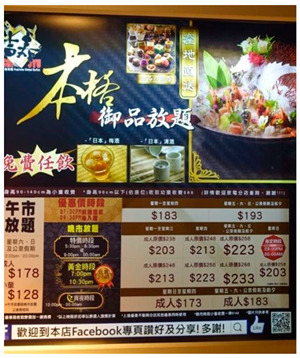
Limited choice of food in the school cafeteria	“There is less vegetable choice in our school canteen. Once I ordered meat rice and there was only one green vegetable in it.” (DCY) 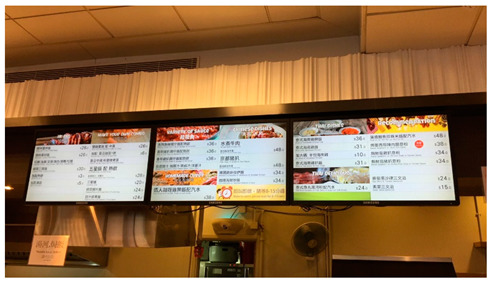
Social norm impact	“I think having a dinner party with friends might make you eat a bit more freely... Because you would be very enjoyable and then order a lot of things, so that can be a barrier to healthy eating.” (WZW)
Affordability and convenience	“This McDonald’s has put its dessert shop outside on the street, which I think is a big obstacle for our health. It’s quite common to pass by and buy an ice cream especially in hot weather like now. And McDonald’s is also cheaper than the others.” (ZZY) 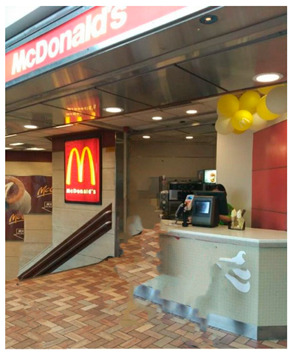

**Table 2 ijerph-19-16558-t002:** Examples of codes and sub-themes on recreational physical activities among university students.

Subthemes	Meaning Units (Codes)
Home and community recreation facilities	“There are three or four sports fields near my home, including soccer and basketball. I will get out and play when I’m free, because I like sports as well.” (FYX) 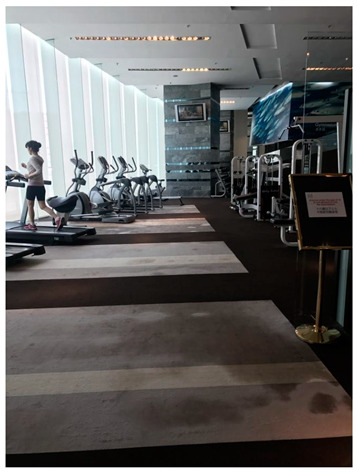
Family support	“(Is there a mutual influence?) Yes, because when I was a child, I did not like to exercise, and then my dad dragged me to run and told me that I was too fat, and I needed to lose weight.” (YWS) 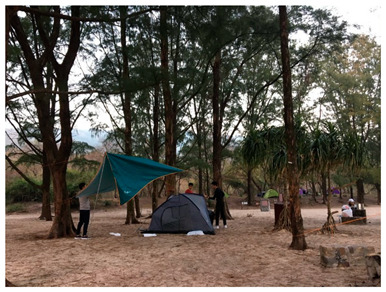
Campus recreation activity	“There are regular activities in the dormitory, and my roommate and I occasionally participate in interhall competitions, such as dodgeball and stickball.” (MK) 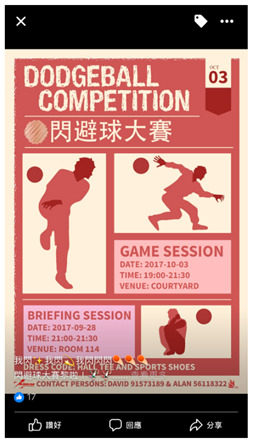
Negative facility condition	“Sometimes it will be renovated, you can also see, is rotten, sealed, people can only pass through that side, it is inconvenient” (YWS) 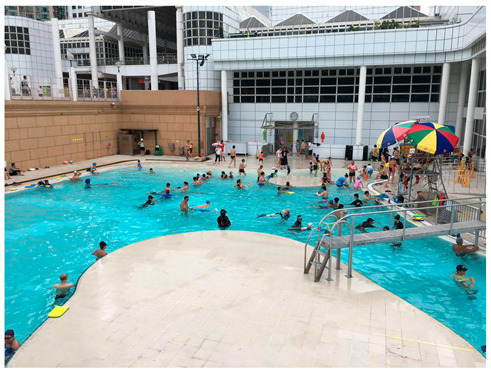
Sport electronic devices	“Using sport electronic devices allows me to understand what my heart rate is like when I’m doing exercise and how many calories I can burn. It gives me more insight into what kind of a result I can get from this exercise, so I like that.” (WZW)

**Table 3 ijerph-19-16558-t003:** Examples of codes and sub-themes on transportation and physical activity among university students.

Subthemes	Meaning Units (Codes)
Convenience of public transportation	“Because the subway is more punctual and will not be jammed, the time is easier to control.” (FXY)
Walking to save money	“No, I mainly consider the fare, that is to say, if I travel by public transport every day, the daily fare will be ten dollars; But if I choose to walk, I save this money and use it for other purposes.” (LMF) 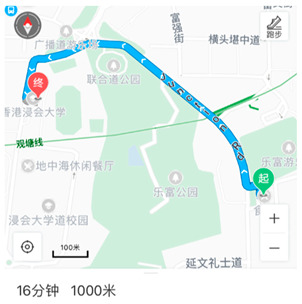
Street condition for walking	“...there will be some cars appearing suddenly next to me, which scares me.” (YWS)“It’s narrow, there is no shade, and it is uncomfortable in the wind and the sun.” (LY) 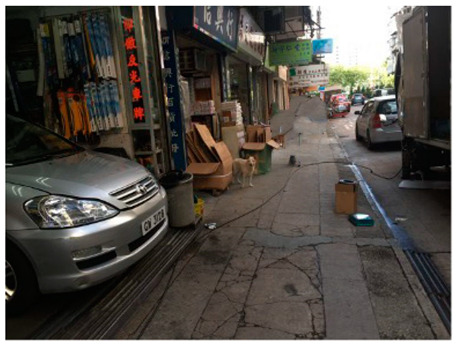 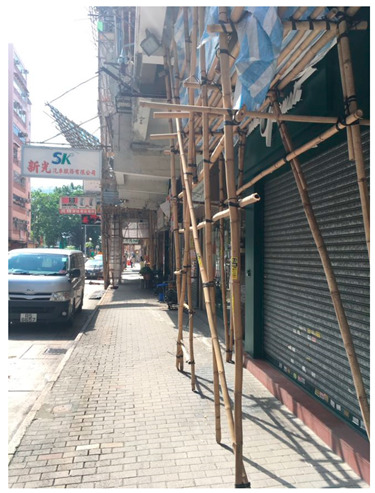 “It’s a relatively quiet road, and there are not many cars passing by. It’s a relatively easy walk to the subway station in about 15 min, and the environment is not bad, and the air is good.” (YXI) 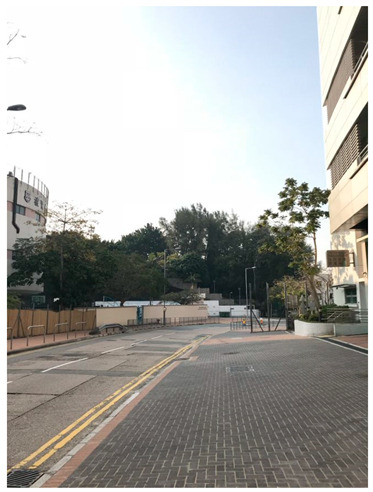
No Cycling Policy	“I think it can be more popular, it is difficult for ordinary people to use bicycles instead of walking.... They may not be able to ride on the road, that is, they are able to use in life.” (CTT) 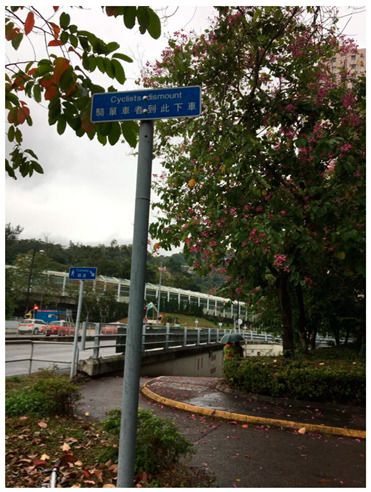

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
