# Peer review of "Neighborhood Built and Social Environment Influences on Lifestyle Behaviors among College Students in a High-Density City: A Photovoice Study"

_ijerph, 2022, doi:10.3390/ijerph192416558_

Round 1

Reviewer 1 Report

This exciting paper used the photovoice method to collect data for in-depth interviews. There are some flaws that need to be corrected:

1. Please characterize the population in more detail.

2. Please consider how representative the examined sample was.

3. What was the agreement between the coders?

4. Have the authors thought about the use of the hermeneutic circle?

5. Please describe the limitations of the study.

6. Please complete the final sections, eg Author Contributions, Institutional Review Board Statement.

7. It is worth presenting the entire code tree in one table to improve readability.

8. Not all elements of COREQ are described in the article.

9. Were questions pilot tested?

10. Were field notes made during and/or after the interview?

11. Was data saturation discussed?

12. Did participants provide feedback on the findings?

13. How were the participants recruited?

Reviewer 2 Report

The first sentence needs to be re-written as the word "being" reads awkwardly. Just state "naturally go through a transitional period..."

Line 38, should be changed to referred "to"...

Line 39-40 their "increased" engagement in activities...

Line 44 that "a" healthy lifestyle..

In the introduction, I would suggest consulting the literature for work done by Svetlana Jovic on green spaces and social narratives to provide a contrast and deeper context to what is presented here. Further, Jovic's work highlights the areas of New York during and continuing through COVID-19 on the positive impacts of green spaces. The introduction lacks discussion on how COVID-19 influenced the need for more engagement in green spaces and outdoor activities of socialization, which is a major problem in the manuscript given how critical this issues still is globally.

The introduction lacks a clear problem statement within the first few sentences of the introduction. It also lacks an understanding of the hypotheses being put forth in the study, which should appear at the end of the introduction leading into the method section.

Methods: (should be written in past tense).

It is a "convenience" sample.

Line 123: bottom-up understanding of the college student's perceptions of healthy behaviors through their photovoice. The next sentence needs to define what actually is a photovoice rather than what they are able to do state what is it exactly.

Line 205-206, an explanation as to why only one set was translated needs to be made clear to the reader.

All faces in photos should be blurred out. Especially, the children as it violates ethics to do so unless you have explicit assent from their parents and consents from the participants. If so, then this information needs to be stated in the Method section.

Results: The results are sound, but are there any statistical differences that can be noted through this approach?

Discussion: There needs to be discussion on the limitation for the isolated area and the convenience sampling that was done. Does this work represent a more generalized distribution of people given the small sample size and the context in which it was done. The main flaw here is that the interview questionnaire is not appended to the manuscript as standard practice. This would allow the reader to repeat and assess for reliability different provinces in China or elsewhere. Further, were pre-post questions asked of the College students on their preference for green spaces and the need to engage in these activities? It is unclear without the questionnaire available. 

Conclusions: Again, it fails to reference impacts of COVID-19 on the topic and how it may be used in the future to leverage mental health and physical health issues given what we have learned about the pandemic.

Reviewer 3 Report

This article examines how university students are affected by the built and social environment in terms of physical activity, transport and food. The article is logical and complete. However, the research methodology is simple and the results are not sufficiently creative. I am sorry that I was unable to pass the review of the article. The specific problems are as follows:

1. Using the photovoice method, 37 subjects were asked to record their environment and then interviewed. The method is not convincing enough for the conclusions. Firstly, the number of subjects was too small and not representative. Secondly, the method of recording was too subjective, as each person recorded through his or her own preference, and there was no comparability between the subjects.

2. The interpretation of the results is too simple. In the results section, the three factors are only interpreted in relation to the subjects, and there are not many findings or summaries.

3. In the discussion section, there are too many references to the literature and no creative findings.

4. The conclusion section is also rather simple. The value of the study is low.

Round 2

Reviewer 2 Report

The method section needs to indicate institutional approval and ethics approval or conducting the research.

The statistical methods, descriptive qualitative or otherwise, need to be explained and references used for which methodology they are replicating and why? Not addressing these matters will place the manuscript in a rejection decision.

Reviewer 3 Report

The author modified the article and made it more complete and clear.But questions about research methods have not been resolved. Although certain research limitations may exist in the research, if the limitations are too large, the results of the research will be appropriate. Therefore, it is recommended to supplement the number of samples.
